# TNCME: TENSOR'S NORM CONSTRAINTS FOR UNSUPERVISED CONTRASTIVE LEARNING OF MULTIMODAL EMBEDDINGS

## ABSTRACT

Multimodal embedding representation has emerged as a hot research topic and has been applied to multimodal retrieval tasks. Unsupervised contrastive learning, represented by InfoNCE, serves as the mainstream training paradigm for multimodal retrieval tasks. However, existing methods generally only optimize the directional alignment of positive pairs in the embedding space, and neglect another fundamental property of the representation tensors: magnitude. Based on this intuitive insight, we propose a **T**ensor's **N**orm **C**onstraints of **M**ultimodal **E**mbeddings framework, TNCME, which focuses on aligning the 2-norm of embedding representations between positive pairs during contrastive learning, jointly trained with the directional alignment pursued by InfoNCE. This approach optimizes the Top-1 performance of visual-language models in multimodal retrieval tasks. We first rigorously prove that the training objective of norm alignment of representations is consistent with the training logic of contrastive learning, and then adapt this objective to multimodal retrieval tasks. Based on the VLM2Vec-V2 framework, we perform training and evaluation across a total of 81 tasks spanning three representative multimodal retrieval categories: Image-Text, VisDoc-Text, and Video-Text. Experimental results demonstrate that the proposed TNCME outperforms baseline methods across all Top-1 metrics. Code open-sourced on anonymously GitHub: `https://anonymous.4open.science/r/TNCME-ICLR/`

## 1 INTRODUCTION

In multimodal retrieval tasks, existing approaches are primarily categorized into two main groups. The first group includes dual-tower architectures, such as CLIP (Radford et al., 2021a), BridgeTower (Xu et al., 2023b), and ManagerTower (Xu et al., 2023a), which encode images and text independently, often using CLIP-ViT and RoBERTa (Liu et al., 2019), and suffer from limited cross-modal interaction and fine-grained semantic alignment. The second group comprises some Vision-Language Models (VLMs) such as LLaVA (Liu et al., 2023a; 2024a;b) and the Qwen-VL series (Bai et al., 2023; Wang et al., 2025b), which adopt a "ViT-Projector-LLM" architecture to inject visual features into large language models (LLMs) via a projector. These models achieve more powerful modal alignment, contextual understanding, and semantic reasoning, making VLM-based representation learning an emerging trend in multimodal retrieval tasks.

Although VLM2Vec(Jiang et al., 2025) and VLM2Vec-V2(Meng et al., 2025) have preliminarily demonstrated the feasibility of applying multimodal retrieval tasks to VLMs, these methods still rely on InfoNCE(van den Oord et al., 2018) as the sole training objective, lacking a refined modeling of the multimodal alignment process. InfoNCE normalizes representations to focus training on directional alignment within a unit hypersphere: bringing positive pairs closer while pushing soft negative pairs more uniform. However, this mechanism has inherent limitations—when semantically mismatched visual-text pairs exhibit similar directions, the model may struggle to distinguish them effectively, thereby weakening its discriminative capability. The issue reveals the limitations of relying solely on directional alignment, prompting us to further focus on another key attribute of representation vectors—magnitude. In this work, we consider the magnitude of semantic representations as their L2 norm. Magnitude reflects the "energy" intensity of embedding vectors in the

feature space, serving as a crucial dimension for characterizing the semantic density and saliency of samples. Unlike cosine similarity, which measures only directional similarity, norm reveals more nuanced distribution characteristics of samples in high-dimensional space. For instance, semantically rich or visually salient images or texts may exhibit larger norms. Thus, imposing norm alignment constraints during training tightens both directional and norm alignment of positive samples, directly boosting Top-1 retrieval performance, as shown in Fig. 1. TNCSE(Zong et al., 2025) improves upon SimCSE(Gao et al., 2021) in semantic textual similarity tasks by introducing a norm alignment constraint on semantic representation tensors, demonstrating the importance of norm alignment in sentence representation learning. Building upon this, we propose integrating norm alignment into a multimodal contrastive learning framework to establish a more comprehensive mechanism for representation alignment. We further propose TNCME, a novel multimodal embedding framework that jointly optimizes directional alignment and norm consistency to enhance positive-pair matching in magnitude while preserving InfoNCE's directional modeling, improving generalization in multimodal retrieval.

To our knowledge, no existing work has conducted a rigorous theoretical analysis of the semantic tensor norm alignment training objective. This objective neither clarifies whether the optimization process aligns with expected logic nor explores the existence of multiple local optima, making it difficult to guarantee convergence stability and alignment consistency. Furthermore, TNCSE is designed for pure text unimodal scenarios, where hidden states encoded by BERT-like models(Devlin et al., 2019; Liu et al., 2019; Reimers & Gurevych, 2019) exhibit slight variation in their norm distributions. In contrast, multimodal tasks involve significant differences in semantic representations between visual and textual from the outset—not only in direction but also in norm scales. Therefore, directly transferring TNCSE's norm alignment objective to multimodal scenarios may lead to training instability or performance degradation. To address these challenges, we first conduct a rigorous theoretical derivation of the training objective for norm alignment, which proves that the designed optimization objective possesses a globally optimal solution, thereby avoiding optimization difficulties caused by multiple local maxima and enhancing the interpretability of the training process. Simultaneously, we verify that the loss function's update direction aligns with the desired norm alignment trend, demonstrating that this mechanism effectively guides positive samples toward consistency in the representation space. Building upon this foundation, we refine the original norm alignment objective to enhance the model's robustness to multimodal feature discrepancies, making it more suitable for the demands of multimodal contrastive learning, which ultimately enables the collaborative optimization of both direction and norm for query-target embedding pairs of positive samples, leading to an improvement of Top-1 retrieval performance.

We implement training and evaluation of the visual-text retrieval task on the Qwen2-VL-2B model with the VLM2Vec-V2 framework. First, we locally reproduce VLM2Vec-Qwen2-VL-2B as the baseline model to ensure consistency in the experimental environment and comparability of results. Subsequently, based on the proposed TNCME framework, we train the improved model, TNCME-Qwen2-VL-2B, and compare it with the baseline under an identical testing environment. Experimental results demonstrate that across 36 image-text retrieval tasks, 27 visdoc-text tasks, and 18 video tasks, TNCME-Qwen2-VL-2B outperforms baseline on multiple key metrics, including Hit, NDCG, Precision, F1, Recall, MAP, and MRR, which fully validates the effectiveness and generalization capability of the proposed method in enhancing multimodal retrieval performance. Furthermore, ablation experiments are conducted to validate the rationality of modifying the tensor norm constraint in the training objective. Visualizations of sample embeddings in two-dimensional space reveal that under norm alignment, the query and target distributions of positive samples converge more closely, indicating that this training objective better aligns with retrieval task requirements.

We summarize the main contributions of this work as follows:

- To our knowledge, we are the first to introduce the concept of norm alignment for semantic representation tensors into multimodal unsupervised contrastive learning, proposing the multimodal embedding framework TNCME and applying it to multimodal retrieval tasks.

- Through rigorous mathematical proof, we demonstrate that the training objective for norm-aligned alignment possesses a unique optimal solution, and we visually illustrate the trend of this loss function.

- Training and evaluation are conducted within the VLM2Vec-V2 framework. We validate the effectiveness of the proposed method across 36 image-text retrieval tasks, 27 VisDoc-

text retrieval tasks, and 18 video-text retrieval tasks. The results demonstrate improvements across all benchmarks in the Top-1 metrics, indicating more significant precision at the highest ranking position.

## 2 RELATED WORKS

Before the ViT-Proj-LLM architecture, CLIP(Radford et al., 2021a) pioneered visual-language alignment using the InfoNCE loss, which enforces directional consistency between image and text embeddings for efficient cross-modal alignment. Subsequent models, such as BLIP(Li et al., 2022) and BLIP-2(Li et al., 2023), introduced cross-attention to enable early multimodal feature fusion, thereby enhancing intermodal interaction and contrastive learning efficacy. BridgeTower(Xu et al., 2023b) and ManagerTower(Xu et al., 2023a) further improved fine-grained alignment via structural optimizations, which still relied primarily on InfoNCE. With the rise of ViT-Proj-LLM architectures, visual-language alignment has transitioned from complex cross-attention to simpler, more efficient feedforward networks. Under this paradigm, Both GME(Zhang et al., 2024) and LamRA(Liu et al., 2025) explore the application of unsupervised contrastive learning in multimodal retrieval tasks. VLM2Vec(Jiang et al., 2025) introduces the MMEB benchmark(Meng et al., 2025) and applies unsupervised contrastive fine-tuning on Phi-3.5-V(Abdin et al., 2024), boosting training efficiency and representation quality via GradCache. VLM2Vec-V2(Meng et al., 2025) introduces a more comprehensive benchmark, MMEB-V2, and employs Qwen2VL(Wang et al., 2025b) as its backbone. According to the VLM2Vec-V2 report, which outperforms mainstream open-source approaches in the MMEB-V2 benchmark.

## 3 METHOD

In this section, we review the tensor norm constraint as a training objective for semantic representations in unsupervised contrastive learning and demonstrate its alignment with the principles of contrastive learning. We then introduce our core method, the multimodal embedding framework TNCME, detailing how it integrates tensor norm constraints with InfoNCE loss for joint training.

### 3.1 REVIEW OF TENSOR NORM CONSTRAINT TRAINING OBJECTIVES

In multimodal retrieval tasks, existing unsupervised contrastive learning methods typically employ InfoNCE loss to learn embeddings by modeling the semantic representation directions of positive and negative sample pairs in hyperspherical space. TNCSE has demonstrated that focusing on the 2-norm of the representation tensor can optimize the performance of BERT-like models in semantic text similarity tasks. Our objective is to prove the effectiveness of this approach and apply it to multimodal retrieval tasks. Therefore, we first briefly review the training objective of the semantic representation tensor 2-norm constraint.

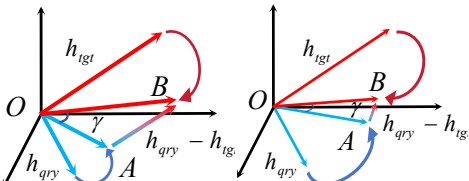

(a) Direction only constrained. (b) Direction and norm are jointly constrained.

Figure 1: These two subfigures illustrate the advantages of norm alignment and direction alignment in three-dimensional space.

If $\mathbf{h}_{qry}$ and $\mathbf{h}_{tgt}$ denote the representations of positive samples in a contrastive learning pair, the tensor norm constraint training objective is defined as Eq. 1:

$$L_{TN}\left(\mathbf{h}_{qry}, \mathbf{h}_{tgt}\right) = \frac{\|\mathbf{h}_{qry} - \mathbf{h}_{tgt}\|}{\|\mathbf{h}_{qry}\| + \|\mathbf{h}_{tgt}\|}, \tag{1}$$

where $\|\cdot\|$ denotes the 2-norm of the tensor. Reducing the high-dimensional semantic representations to a three-dimensional space for visualization, as shown in Fig. 1, it is evident that we can perform vector subtraction on $\mathbf{h}_{qry}$ and $\mathbf{h}_{tgt}$. Since all tensors in Eq. 1 have been normalized to the 2-norm, we obtain a triangle $\triangle OAB$. Expanding the numerator using the cosine theorem gives Eq. 2:

$$L_{TN}\left(\mathbf{h}_{qry}, \mathbf{h}_{tgt}\right) = \frac{\sqrt{\|\mathbf{h}_{qry}\|^2 + \|\mathbf{h}_{tgt}\|^2 - 2\|\mathbf{h}_{qry}\|\|\mathbf{h}_{tgt}\|\cos\gamma}}{\|\mathbf{h}_{qry}\| + \|\mathbf{h}_{tgt}\|}, \tag{2}$$

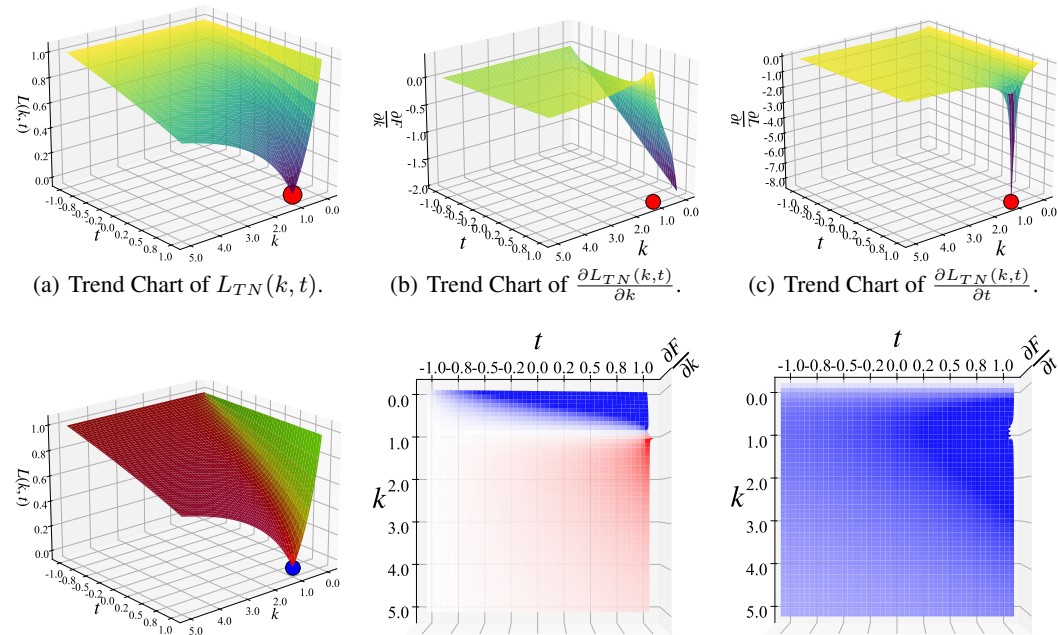

(a) Trend Chart of $L_{TN}(k,t)$.

(b) Trend Chart of $\frac{\partial L_{TN}(k,t)}{\partial k}$.

(c) Trend Chart of $\frac{\partial L_{TN}(k,t)}{\partial t}$.

(d) The direction and magnitude of the gradient of $L_{TN}(k,t)$.

(e) View from above to observe the sign and magnitude of $\frac{\partial L_{TN}(k,t)}{\partial k}$.

(f) View from above to observe the sign and magnitude of $\frac{\partial L_{TN}(k,t)}{\partial t}$.

Figure 2: The series of subfigures quantitatively analyzes $L_{TN}$. Subfigures (a)-(c) show trends of its primitive function and partial derivatives in $t$ and $k$. Subfigure (d) overlays a gradient field on the function surface, red and green areas mark decreases driven by $t$ and $k$, respectively, with darker shades indicating stronger influence. Subfigures (e) and (f) provide top-down views of sign distributions for $\frac{\partial L_{TN}}{\partial k}$ and $\frac{\partial L_{TN}}{\partial t}$, where blue/red denote negative/positive gradients, and darker colors show larger absolute values. For clarity, we mark the point $(1, 1)$ in subfigures (a)-(d).

here, $\gamma$ denotes the angle between tensors $\mathbf{h}_{qry}$ and $\mathbf{h}_{tgt}$. Since it is impossible to explicitly express the relationship between $\mathbf{h}_{qry}$ and $\mathbf{h}_{tgt}$ in any pair of positive samples, without loss of generality, we set $\|\mathbf{h}_{tgt}\| = k \cdot \|\mathbf{h}_{qry}\|$, $k \in (0, +\infty)$ and $t = \cos\gamma$, $t \in [-1, 1]$. Thus, Eq. 2 can be rewritten as a bivariate function of $k$ and $t$, as shown in Eq. 3, which is visualized as Fig 2(a).

$$L_{TN}(k,t) = \frac{\sqrt{1 + k^2 - 2 \cdot kt}}{1 + k}. \tag{3}$$

The ideal objective for $L_{TN}$ is to simultaneously satisfy $\|\mathbf{h}_{qry}\| = \|\mathbf{h}_{tgt}\|$ and $\cos\gamma = 1$. This objective precisely corresponds to $k = 1$ and $t = 1$, ensuring perfect alignment between the two in both norm and direction. Although this configuration is intuitively desirable, existing works have not provided quantitative convergence proofs for this objective, nor has it systematically explored whether other local optima exist.

### 3.2 PROOF OF MONOTONICITY FOR TENSOR NORM CONSTRAINED LOSS FUNCTION $L_{TN}$

While TNCSE empirically shows that norm constraints on semantic representation tensors improve BERT-like models in semantic similarity tasks, it lacks theoretical justification. Here, we rigorously analyze why norm constraints benefit contrastive learning.

First, note that the Eq. 3 can be made a simple transformation. For any $k \in (0, +\infty)$ and $t \in [-1, 1]$, we have Ineq. 4:

$$0 \le \frac{|k - 1|}{k + 1} = \frac{\sqrt{1 + k^2 - 2 \cdot k}}{1 + k} \le \frac{\sqrt{1 + k^2 - 2 \cdot kt}}{1 + k} \le \frac{\sqrt{1 + k^2 + 2 \cdot k}}{1 + k} = 1, \tag{4}$$

which implies that $L_{TN}(k,t) \ge 0$ throughout training, ensuring stable gradient updates. We expect $k$ and $t$ to converge to 1, driving $h_{qry}$ and $h_{tgt}$ to align in both norm and direction, which is consistent with our design motivation. To guarantee reliable optimization, however, we must formally

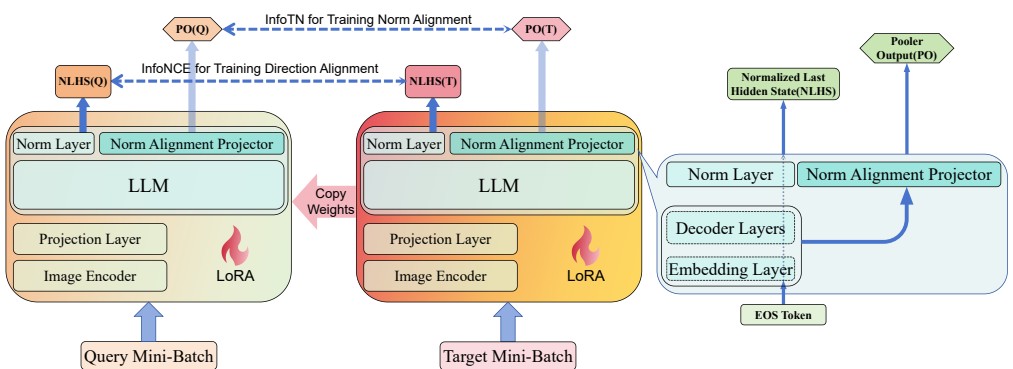

Figure 3: The architecture of our multimodal embedding representation framework, TNCME, which adds an FFN (Norm Alignment Projector, NAP) to a VLM. The hidden states are output by the LLM within the VLM, which passes through both a Norm Layer and the NAP. The resulting normalized last hidden state and projector output are trained by InfoNCE and InfoTN, respectively, to align the direction and norm of the multimodal query and target embedding representations.

show that $k$ and $t$ monotonically approach 1 without being trapped in spurious local optima. In the following, we prove that $L_{\text{TN}}(k,t)$ is monotonic in both $k$ and $t$.

First, we take the partial derivative of $L_{TN}(k,t)$ with the independent variable $t$ to obtain Eq. 5:

$$\frac{\partial L_{TN}(k,t)}{\partial t} = -\frac{k}{(1+k)\cdot\sqrt{k^2 - 2kt + 1}}, \tag{5}$$

It is observed that $\frac{\partial L_{TN}(k,t)}{\partial t} < 0$ holds throughout its domain[1]. Therefore, $L_{TN}(k,t)$ is monotonically decreasing in the $t$-direction, which means that for any $k > 0$ and $k \neq 1$, $L_{TN}(k,t)$ attains its minimum value at $t = 1$, and we visualize $\frac{\partial L_{TN}(k,t)}{\partial t}$ in Fig. 2(c) and 2(f). Then, we take the partial derivative of $L_{TN}(k,t)$ with respect to $k$, obtaining Eq. 6:

$$\frac{\partial L_{TN}(k,t)}{\partial k} = \frac{(k-1)\cdot(1+t)}{(1+k)^2\cdot\sqrt{k^2 - 2kt + 1}}. \tag{6}$$

For any $t \in [-1, 1)$, the behavior of $L_{\text{TN}}(k,t)$ with respect to $k$ is as follows: when $k \in [0,1)$, $\frac{\partial L_{\text{TN}}(k,t)}{\partial k} < 0$, indicating that $L_{\text{TN}}$ is decreasing in $k$; when $k > 1$, $\frac{\partial L_{\text{TN}}(k,t)}{\partial k} > 0$, indicating that $L_{\text{TN}}$ is increasing in $k$. Thus, $L_{\text{TN}}(k,t)$ attains a local minimum at $k = 1$ along the $k$-direction. The behavior of $\frac{\partial L_{\text{TN}}(k,t)}{\partial k}$ is visualized in Figs. 2(b) and 2(e).

In summary, we rigorously prove that $L_{\text{TN}}(k,t)$ has a unique global minimum at $(k,t) = (1,1)$ within its domain, where $L_{\text{TN}}(k,t) = 0$. This corresponds to perfect alignment between query and target in both norm and direction of their semantic representations, precisely the objective of our training framework. To further illustrate the optimization trajectory of $L_{\text{TN}}(k,t)$, Fig. 2(d) visualizes the dominant influence of each variable during descent. The surface is color-mapped according to the gradient's direction and magnitude, highlighting the steepest descent directions across the domain. This reveals how $k$ and $t$ jointly govern the functional landscape and drive convergence toward the global minimum.

## 3.3 MODEL STRUCTURE DESIGN

We employ a Qwen2VL model to encode queries and targets, utilizing the encoded last hidden states (LHS) for training. We observe that the LLM output representation tensor in the QwenVL series models will be normalized by RMSNorm(Zhang & Sennrich, 2019) to obtain the LHS. The LHS is clipped of norm features, retaining only directional features. VLM2Vec-V2 employs InfoNCE to set directional constraints on the LHS, and we need an FFN to reconstruct the norm features. Inspired by TNCSE, we observe that during generative tasks, QwenVL's LHS also passes through an FFN, referred to as the language model head (LM head). This head maps the LHS into a tensor of the

---

[1]We discuss in detail in Appendix A the effect of discontinuities that cause the denominator to become zero on monotonicity.

vocabulary size dimension, defined as logits, which are used for autoregressive tasks. Intuitively, one might consider using logits for tensor norm constraints. However, we find the logit distribution is excessively sparse and the output dimension of the LM head is prohibitively large, causing memory overflow even with LoRA(Hu et al., 2022) fine-tuning. Thus, we define a randomly initialized FFN after the LLM, as the **Norm Alignment Projector (NAP)**. Its purpose is to guide the model toward focusing on the semantic features required for retrieval tasks. The input and output dimensions of NAP align with the LHS, preventing memory overflow. We denote the features obtained by passing LHS through NAP as **Projector Output (PO)**, which captures the norm features of the representation. We employ PO in $L_{TN}$ and LHS in InfoNCE, and train them jointly, as Fig. 3. NAP is used only in training, thus, the inference pipelines of TNCME and VLM2Vec-V2 are identical.

Given TNCSE's focus on pure text modalities, $L_{TN}$ can be directly utilized for intuitive constraints. However, since TNCME focuses on multimodal data representation, ablation experiments reveal that directly applying $L_{TN}$ results in convergence issues for the loss. Therefore, this subsection modifications to the $L_{TN}$ is outlined in subsection 3.1, aiming to enhance its suitability for unsupervised contrastive learning of multimodal embeddings. First, we present the InfoNCE, in Eq. 7:

$$L_{InfoNCE} = -\log \frac{e^{sim(\mathbf{h}_{qry}, \mathbf{h}_{tgt+})/\tau}}{e^{sim(\mathbf{h}_{qry}, \mathbf{h}_{tgt+})/\tau} + \sum_{tgt- \in \mathbb{N}} e^{sim(\mathbf{h}_{qry}, \mathbf{h}_{tgt-})/\tau}}, \tag{7}$$

where $sim$ denotes cosine similarity, $\tau$ denotes the temperature coefficient, and $\mathbb{N}$ denotes the current mini-batch being trained. We observe that in the vast majority of cases, the cosine similarity distribution for multimodal positive and negative sample embeddings ranges from -0.01 to 1. According to Eq. 4, we have $0 \leq L(k,t) \leq 1$. Therefore, the range distribution of $L_{TN}(k,t)$ approximates the actual cosine similarity. Consequently, we define the norm similarity, as Eq. 8:

$$sim_{TN} = 1 - L_{TN}. \tag{8}$$

Since $L_{TN}(k,t)$ is expected to decrease during training while $sim_{TN}$ increases, and $0 \leq sim_{TN} \leq 1$, we intuitively combine $sim_{TN}$ and InfoNCE to propose the contrastive learning objective $Info_{TN}$ for multimodal embeddings, defined as Eq. 9:

$$L_{InfoTN} = -\log \frac{e^{sim_{TN}(\mathbf{h}_{qry}, \mathbf{h}_{tgt+})/\tau_{TN}}}{e^{sim_{TN}(\mathbf{h}_{qry}, \mathbf{h}_{tgt+})/\tau_{TN}} + \sum_{tgt- \in \mathbb{N}} e^{sim_{TN}(\mathbf{h}_{qry}, \mathbf{h}_{tgt-})/\tau_{TN}}}, \tag{9}$$

where $\tau_{TN}$ is also a temperature coefficient, independent of $\tau$. This design effectively mitigates optimization obstacles caused by excessive differences in norm across different modalities, avoiding abrupt changes in the loss function triggered by the forced alignment of norm, and promotes stable convergence of the loss during training. We will visualize the changes in $L_{InfoTN}$ and $L_{TN}$ during training in ablation experiments to demonstrate the smoothness of InfoTN. Ultimately, we combine InfoTN and InfoNCE through joint training, defining the overall loss function as Eq. 10.

$$L = \lambda \cdot L_{InfoNCE} + (1 - \lambda) \cdot L_{InfoTN}, \lambda \in (0, 1). \tag{10}$$

# 4 EXPERIMENTS

## 4.1 EXPERIMENTAL SETUP

We conduct experiments based on the VLM2Vec-V2 framework[2], which consists of three training tasks: Image-Text retrieval, VisDoc-Text retrieval, and Video-Text retrieval. All data are sourced from MMEB-train(Meng et al., 2025). To validate the method's generalization capability, we design three sets of progressive experiments: **(i)** Training solely on Image-Text data to evaluate image-text retrieval performance, and evaluating VisDoc-Text and Video-Text retrieval tasks under zero-shot conditions; **(ii)** Jointly training Image-Text and VisDoc-Text data, evaluating performance on both tasks, and conducting zero-shot evaluation on the Video-Text retrieval task; **(iii)** We use the full training set and then evaluate on three-class tasks[3]. Details of the datasets are listed in Appendix C. To thoroughly evaluate performance, we use multiple metrics,

---

[2]https://github.com/TIGER-AI-Lab/VLM2Vec

[3]Since the performance of reproduced VLM2Vec-V2 on the all training set shows a decline compared to the reported results, the issue remains unresolved as the manuscript submission; details are in Appendix B.

Table 1: We evaluate VLM2Vec-V2-Qwen2VL and TMCSE-Qwen2VL on three multimodal retrieval tasks: Image-Text (Im, 36 items), VisDoc-Text (Vd, 27 items), and Video-Text (Vi, 18 items). Results are averaged across subtasks and reported for eight metrics: Hit@k (H@k), NDCG-Linear@k (NL@k), NDCG-Exponential@k (NE@k), Precision@k (P@k), Recall@k (R@k), F1@k, MAP@k (MA@k), and MRR@k (MR@k). Top-1 scores are visually highlighted in ▨. We also report the average improvement of TNCME over VLM2Vec-V2 separately, both the overall average (**Avg**) and the average @1 (**Avg @1**) derived from a total of 81 tasks.

| Model | VLM2Vec-V2 | | | TNCME | | | VLM2Vec-V2 | | | TNCME | | | VLM2Vec-V2 | | | TNCME | | |
|---|---|---|---|---|---|---|---|---|---|---|---|---|---|---|---|---|---|---|
| Training | Image Only (5000 Steps) | | | | | | Image and VisDoc (5000 Steps) | | | | | | All Training-sets (2000 Steps) | | | | | |
| Metric | Im | Vd | Vi | Im | Vd | Vi | Im | Vd | Vi | Im | Vd | Vi | Im | Vd | Vi | Im | Vd | Vi |
| | 36 | 27 | 18 | 36 | 27 | 18 | 36 | 27 | 18 | 36 | 27 | 18 | 36 | 27 | 18 | 36 | 27 | 18 |
| H@1 | 63.7 | 22.9 | 30.0 | **64.9** | **24.5** | **31.4** | 64.5 | 47.9 | 32.9 | **65.3** | **49.1** | **33.0** | 62.4 | 49.4 | 35.0 | **62.8** | **50.3** | **35.2** |
| H@5 | 84.1 | 42.5 | 70.3 | **85.0** | **43.0** | **72.1** | 84.5 | 71.1 | **73.2** | **84.9** | **72.2** | 72.9 | 83.9 | 73.0 | 74.1 | 83.9 | 72.2 | 74.1 |
| H@10 | 88.7 | **52.1** | 82.0 | **89.5** | 52.1 | **83.1** | 89.3 | 78.6 | **83.6** | **89.5** | **80.1** | 83.2 | 88.7 | 79.6 | 84.3 | 88.7 | 79.1 | 84.3 |
| NL@1 | 63.7 | 21.6 | 30.0 | **64.9** | **23.3** | **31.4** | 64.5 | 46.2 | 32.9 | **65.3** | **47.4** | **33.0** | 62.4 | 47.8 | 35.0 | **62.8** | **48.6** | **35.2** |
| NL@5 | 75.0 | 28.5 | 51.3 | **76.1** | **29.7** | **52.8** | 75.6 | 54.7 | **54.2** | **76.2** | **56.0** | **54.2** | 74.3 | 56.2 | 55.7 | **74.5** | **56.3** | **55.9** |
| NL@10 | 76.5 | 30.9 | 55.1 | **77.6** | **32.0** | **56.4** | 77.1 | 56.7 | 57.5 | **77.7** | **58.3** | **57.6** | 75.9 | 57.9 | 59.0 | **76.1** | **58.0** | **59.2** |
| NE@1 | 63.7 | 20.7 | 30.0 | **64.9** | **22.5** | **31.4** | 64.5 | 45.1 | 32.9 | **65.3** | **46.2** | **33.0** | 62.4 | 46.7 | 35.0 | **62.8** | **47.4** | **35.2** |
| NE@5 | 75.0 | 28.0 | 51.3 | **76.1** | **29.3** | **52.8** | 75.6 | 54.0 | **54.2** | **76.2** | **55.4** | **54.2** | 74.3 | 55.6 | 55.7 | **74.5** | **55.7** | **55.9** |
| NE@10 | 76.5 | 30.6 | 55.1 | **77.6** | **31.7** | **56.4** | 77.1 | 56.3 | 57.5 | **77.7** | **57.9** | **57.6** | 75.9 | 57.5 | 59.0 | **76.1** | **57.6** | **59.2** |
| P@1 | 63.7 | 22.9 | 30.0 | **64.9** | **24.5** | **31.4** | 64.5 | 47.9 | 32.9 | **65.3** | **49.1** | **33.0** | 62.4 | 49.4 | 35.0 | **62.8** | **50.3** | **35.2** |
| P@5 | 16.8 | **11.9** | 14.1 | **17.0** | 11.4 | **14.5** | 16.9 | 19.9 | **14.7** | **17.0** | 20.5 | **14.7** | 16.8 | 20.5 | 14.9 | 16.8 | 20.5 | 14.9 |
| P@10 | 8.9 | **8.8** | 8.3 | **9.0** | 8.1 | **8.4** | 8.9 | 13.7 | **8.4** | **9.0** | 14.1 | **8.4** | 8.9 | 13.6 | 8.5 | 8.9 | 13.6 | 8.5 |
| R@1 | 63.7 | 16.2 | 29.9 | **64.9** | **18.3** | **31.3** | 64.5 | 36.9 | 32.8 | **65.3** | **38.3** | **32.9** | 62.4 | 38.2 | 34.8 | **62.8** | **38.6** | **35.1** |
| R@5 | 84.1 | 31.5 | 70.2 | **85.0** | **32.9** | **72.0** | 84.5 | 57.5 | **73.2** | **84.9** | **58.5** | 72.8 | 83.9 | 59.1 | 74.0 | 83.9 | 58.6 | **74.1** |
| R@10 | 88.7 | 40.2 | 82.0 | **89.5** | **40.9** | **83.1** | 89.3 | 65.7 | **83.6** | **89.5** | **67.4** | 83.2 | 88.7 | 66.6 | 84.3 | 88.7 | 66.2 | 84.3 |
| F@1 | 63.7 | 16.8 | 29.9 | **64.9** | **18.9** | **31.3** | 64.5 | 38.1 | **32.9** | **65.3** | **39.5** | 32.9 | 62.4 | 39.5 | 34.9 | **62.8** | **40.0** | **35.1** |
| F@5 | 28.1 | 13.0 | 23.5 | **28.3** | **13.2** | **24.1** | 28.2 | 23.2 | 24.5 | **28.3** | **23.8** | 24.4 | 28.0 | 23.9 | 24.8 | **28.0** | 23.9 | 24.8 |
| F@10 | 16.1 | **10.7** | 15.0 | **16.3** | 10.3 | **15.2** | 16.2 | 17.2 | **15.3** | **16.3** | **17.7** | 15.2 | 16.1 | 17.3 | 15.4 | 16.1 | 17.2 | 15.4 |
| MA@1 | 63.7 | 22.9 | 30.0 | **64.9** | **24.5** | **31.4** | 64.5 | 47.9 | 32.9 | **65.3** | **49.1** | **33.0** | 62.4 | 49.4 | 35.0 | 62.6 | **50.3** | **35.2** |
| MA@5 | 72.0 | 25.2 | 44.9 | **73.1** | **26.7** | **46.4** | 72.6 | 50.5 | 47.8 | **73.2** | **52.1** | **48.0** | 71.1 | 51.9 | 49.5 | **71.3** | **52.2** | **49.9** |
| MA@10 | 72.6 | 25.6 | 46.5 | **73.7** | **27.0** | **48.0** | 73.2 | 50.6 | 49.2 | **73.9** | **52.3** | **49.4** | 71.7 | 51.7 | 50.9 | **72.0** | **52.0** | **51.1** |
| MR@1 | 63.7 | 22.9 | 30.0 | **64.9** | **24.5** | **31.4** | 64.5 | 47.9 | 32.9 | **65.3** | **49.1** | **33.0** | 62.4 | 49.4 | 35.0 | **62.8** | **50.2** | **35.2** |
| MR@5 | 72.0 | 29.9 | 45.0 | **73.1** | **31.1** | **46.5** | 72.6 | 56.7 | 47.9 | **73.2** | **57.9** | **48.0** | 71.1 | 58.5 | 49.6 | **71.3** | **58.8** | **49.9** |
| MR@10 | 72.6 | 31.2 | 46.7 | **73.7** | **32.3** | **48.0** | 73.2 | 57.7 | 49.3 | **73.9** | **59.0** | **49.4** | 71.7 | 59.4 | 50.9 | **72.0** | **59.7** | **51.2** |
| Avg | 63.2 | 25.3 | 41.7 | **64.1** | **26.4** | **42.9** | 63.8 | 47.6 | **44.1** | **64.3** | **48.8** | 44.0 | 62.5 | 48.8 | 45.4 | **62.7** | **49.0** | **45.6** |
| Avg @1 | | 41.9 | | | **43.3(+1.4%)** | | | 50.9 | | | **51.7(+0.8%)** | | | 50.9 | | | **51.4(+0.5%)** | |

including Hit@1,5,10, NDCG-linear@1,5,10, NDCG-exponential@1,5,10, Precision@1,5,10, Recall@1,5,10, F1@1,5,10, MAP@1,5,10, and MRR@1,5,10, which capture retrieval quality from complementary perspectives as summarized in Appendix D[4]. Consistent with VLM2Vec-V2, we employ Qwen2-VL-2B as the backbone, and use the EOS token as the pooling method. All training is completed on 8 H100 GPUs with a total batch size set to 1024, a learning rate set to 5e-5, and a linear decay strategy is adopted. We adopt the LoRA fine-tuning strategy with rank set to 16 and scaling factor $\alpha$ set to 64. LoRA is implemented based on the PEFT(Mangrulkar et al., 2022) framework. In the first two experiments, which are conducted on subsets of the training data, we train for 5,000 steps. Due to computational resource constraints, we reduce the training steps to 2,000 when using the full training set[5]. The InfoNCE temperature $\tau$ is kept at its default value of 0.02 and $\lambda$ is fixed at 0.5 across all configurations. To account for differences in training set distribution, we set $\tau_{TN}$ to $10^{-4}$ for the full training set, while the subset-based experiments use $\tau_{TN} = 0.05$.

## 4.2 BASELINE SETTING

VLM2Vec-V2 has demonstrated that VLM2Vec-V2-Qwen2VL-2B outperforms several recently open-sourced visual-language retrieval baselines across Image-Text, Video-Text, and Visdoc-Text tasks, such as ColPali-v1.3(Faysse et al., 2025a), GME-2B/7B(Zhang et al., 2024), LamRA-

---

[4]VLM2Vec-V2 employs Hit@1 as the metric for image-text retrieval tasks and VisDoc-text retrieval tasks, while NDCG@5 is used for video-text retrieval tasks. To comprehensively evaluate the model's performance across different tasks, we utilize all available evaluation metrics for a comprehensive evaluation.

[5]In VLM2Vec-V2, the authors mention training for either 2K or 5K steps; our settings are thus aligned with the original paper.

Qwen2/2.5-VL(Liu et al., 2025), VLM2Vec-2B/7B(Jiang et al., 2025), etc, as summerized in Appendix B. To demonstrate the superiority of our proposed method over VLM2Vec-V2, we retrain VLM2Vec-V2-Qwen2VL-2B as a baseline comparison using the same GPUs, official source code, and default parameters under three experimental conditions.

## 4.3 RESULTS ANALYSIS

We report the evaluation results of three experimental sets in Table 1. The experiments fully validate TNCME's significant advantage in enhancing Top-1 retrieval performance: across three distinct training set configurations, TNCME consistently outperforms the baseline model VLM2Vec-V2 on Top-1 metrics for all multimodal retrieval tasks (Image-Text, VisDoc-Text, Video-Text), demonstrating consistent and cross-modal generalization capabilities. Even when trained solely on image-text data, regardless of whether VisDoc or Video data is introduced, TNCME consistently maintains its lead, which demonstrates that the norm alignment mechanism exhibits strong robustness to training data composition, does not rely on specific modality distributions, and possesses broad applicability. Notably, even under the All-training-sets setting with only 2000 training steps, TNCME maintains its lead in Top-1 metrics, further demonstrating its stability. Although some metrics exhibit fluctuations at @5 and @10, which reflects a reasonable trade-off made to prioritize first-hit accuracy and does not undermine the model's core strengths.

## 5 ANALYSIS AND ABLATION STUDY

### 5.1 EMBEDDING SPACE ANALYSIS

To more intuitively demonstrate the multimodal alignment between query and target embeddings, we employ t-SNE(Cieslak et al., 2020) visualization to analyze the embedding spaces of Qwen2VL, VLM2Vec-V2-Qwen2VL, and TNCME-Qwen2VL. Specifically, we randomly select 100 identical query-target sample pairs from the test set, reduce their high-dimensional embeddings to a two-dimensional space using t-SNE, and compute the Euclidean distance between each pair in this space. To enhance visualization, we connect queries and targets that are positive samples with each other using gray lines, as shown in Fig. 4. Visualization results indicate that the original Qwen2VL, untrained with contrastive learning, exhibits highly dispersed distributions of image and text

Table 2: This table reports ablation results about whether to add FFN and whether to combine the training objective with InfoNCE, comparing with VLM2Vec-V2 and TNCME. All experiments are completed on image-text retrieval tasks.

| Method | 36 Avg Hit@1 |
|---|---|
| VLM2Vec-V2 | 63.7 |
| w/o. NAP | 62.4 |
| InfoNCE+$L_{TN}$ | 63.8 |
| **Our setting** | **64.9** |

embeddings, reflecting a lack of alignment between modalities. After introducing InfoNCE for directional alignment in VLM2Vec-V2-Qwen2VL, the distribution of positive pairs converges significantly, though some misalignment persists between modalities. TNCME-Qwen2VL achieves tighter positive sample clustering in the embedding space by jointly optimizing directional and norm alignment, significantly enhancing multimodal consistency. We quantitatively measure alignment performance by labeling the average Euclidean distance of the corresponding model in the 2D space on 100 sample pairs for each subgraph. Experimental, results show that compared to Qwen2VL and VLM2Vec-V2-Qwen2VL, TNCME-Qwen2VL reduces the average distance by 89.24% and 46.18%, respectively, fully validating the proposed method's significant advantage in enhancing cross-modal retrieval alignment capabilities for Qwen2VL.

### 5.2 WHY NOT USE $L_{TN}$ DIRECTLY?

Fig. 4 illustrates a significant semantic gap between visual and textual representations in the backbone's initial state. Due to the batch size of 1024 during training, substantial discrepancies emerged in the norm of query and target representations, causing the $L_{TN}$ loss to fluctuate violently during optimization and hindering convergence. This phenomenon doesn't occur in the pure text-modal TNCSE, indicating that the variance in representation norm distributions poses a challenge to training stability in multimodal scenarios; thus, the original method performs ineffectively. To address this, we construct InfoTN by combining $L_{TN}$ with InfoNCE, which effectively mitigates the convergence issues caused by the variance in multimodal representation norm distributions, leading to a smoother and more stable loss curve. Fig. 5(b) compares the overall loss trends under both strate-

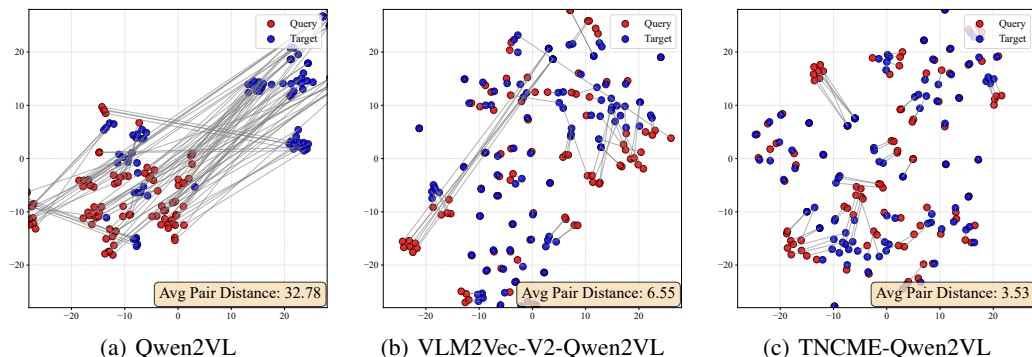

(a) Qwen2VL      (b) VLM2Vec-V2-Qwen2VL      (c) TNCME-Qwen2VL

Figure 4: This series of subfigures visualizes the embedding distributions of Qwen2VL, VLM2Vec-V2-Qwen2VL, and TNCME-Qwen2VL across 100 random sample pairs in a 2-dimensional space.

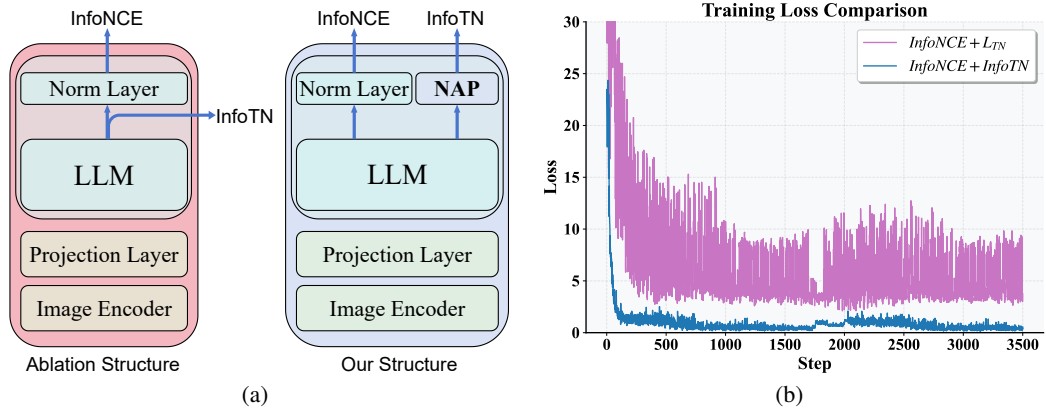

Figure 5: Subfigure (a) illustrates the difference in training architecture with and without the NAP; Subfigure (b) reports that the loss fails to converge when training directly with $L_{TN}$ in the image-text retrieval task.

gies, while Table 2 reports the final training performance results, confirming the proposed method's advantages in convergence and effectiveness.

## 5.3 WHY USE AN EXTERNAL FFN (NAP)?

The purpose of adding a feedforward neural network **NAP** in TNCME is to reconstruct the norm feature of the normalized last hidden state for joint training. Since the LLM output representation in Qwen2VL is not normalized and inherently possesses norm features, this representation passes through a final RMSNorm for normalization. The resulting LHS is then utilized for unsupervised contrastive learning training. To obtain the norm feature, an intuitive approach is to use the hidden state input to InfoTN without RMSNorm for norm alignment, while employing LHS for InfoNCE to achieve directional alignment, as shown in Fig. 5(a). However, this may be influenced by generative pretraining, introducing noise unrelated to norm alignment. Therefore, our approach involves initializing a decoupled FNN that maps raw hidden states to a new representation space, making their norm features more suitable for the InfoTN loss. We evaluate on 36 image-text retrieval tasks with consistent training hyperparameters, reporting results in Table 2.

## 6 CONCLUSION

In this paper, we first prove that norm alignment for embeddings is theoretically consistent with contrastive learning objectives. Building on this, we adapt the norm alignment objective for multimodal retrieval, aiming to boost Top-1 performance across metrics. Based on the VLM2Vec-V2 framework, we propose TNCME, a novel embedding approach trained with Qwen2VL-2B as the backbone. Evaluated on image-text, VisDoc-text, and video-text retrieval tasks, TNCME-Qwen2VL-2B consistently outperforms the replicated VLM2Vec-V2 baseline across all metrics. Ablation studies further validate the effectiveness of our method.

FUTURE WORKS

We adopt the latest Qwen2.5-VL(Bai et al., 2025) as both our backbone and the backbone for VLM2Vec-V2; however, experimental results show that it underperforms Qwen2-VL. This may stem from Qwen2.5VL employing an overly flexible dynamic pixel scaling strategy. Under large batch sizes, it necessitates compressing maximum pixel values to prevent out-of-memory errors. This may limit Qwen2.5VL's semantic expression capabilities, leading to suboptimal results. In the future, we will explore multimodal representation learning methods for VLM with flexible dynamic pixel scaling, exemplified by Qwen2.5-VL.

THE USAGE OF LLM

In this work, we use LLM to polish the mathematical derivation subsection in Appendix A and polish the Introduction and Method sections.

REPRODUCIBILITY STATEMENT

We have open-sourced the training and evaluation code for TNCME on an anonymous GitHub repository. Key hyperparameters are detailed in the Experimental Setup section of the paper. For further details, please refer to the training code.

ETHICS STATEMENT

This study does not involve any personal data, sensitive information, or high-risk application scenarios. No ethically controversial datasets or models were used. All experimental data are drawn from publicly available multimodal benchmark datasets, and the sole purpose of this research is to advance the development of multimodal representation learning. The study adheres strictly to data usage guidelines and does not involve any processing of the original data that could raise privacy or bias concerns. Therefore, we believe this work poses no significant ethical risks.

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

# A  ANALYSIS OF PARAMETERS FOR $\frac{\partial L_{TN}(k,t)}{\partial t}$

The partial derivative of $L_{TN}(k,t)$ with respect to $t$ is given by

$$\frac{\partial L_{TN}(k,t)}{\partial t} = -\frac{k}{1+k} \cdot \frac{1}{\sqrt{1+k^2-2kt}}, \tag{A-1}$$

where the domain is defined as $k \in [0,+\infty)$ and $t \in [-1,1]$. Observe that for all $k > 0$, the prefactor $-\frac{k}{1+k} < 0$, and at $k = 0$, the derivative vanishes (a trivial case). Thus, the sign of $\frac{\partial L_{TN}(k,t)}{\partial t}$ is entirely governed by the term:

$$h(k,t) = \frac{1}{\sqrt{f(k,t)}}, \quad \text{where} \quad f(k,t) = 1 + k^2 - 2kt. \tag{A-2}$$

Since $h(k,t)$ involves a square root in the denominator, we require $f(k,t) > 0$ for well-definedness. We now analyze where $f(k,t) = 0$ within the domain $k \geq 0, \ t \in [-1,1]$.

Treat $f(k,t)$ as a quadratic in $k$ with parameter $t \in [-1,1]$:

$$f(k,t) = k^2 - 2tk + 1.$$

Its discriminant is

$$\Delta = (-2t)^2 - 4 \cdot 1 \cdot 1 = 4(t^2 - 1).$$

We consider the following cases:

- **Case 1:** $\Delta > 0$, i.e., $|t| > 1$. This implies two distinct real roots in $k$, but such values of $t$ lie outside the domain $[-1,1]$. Hence, no solutions exist in the feasible region.

- **Case 2:** $\Delta = 0$, i.e., $t = \pm 1$.
  - If $t = 1$, then $f(k,1) = (k-1)^2$, which vanishes when $k = 1$. Thus, $(k,t) = (1,1)$ is a zero of $f(k,t)$.
  - If $t = -1$, then $f(k,-1) = (k+1)^2$, which vanishes when $k = -1$. However, since $k \geq 0$, this point lies outside the domain and is discarded.

- **Case 3:** $\Delta < 0$, i.e., $|t| < 1$. Then $f(k,t) > 0$ for all $k \in [0,+\infty)$, meaning no real roots exist and the expression under the square root remains strictly positive.

Therefore, the only point in the domain where $f(k,t) = 0$ is $(k,t) = (1,1)$. Consequently,

- $f(k,t) > 0$ for all $(k,t) \in [0,+\infty) \times [-1,1] \setminus \{(1,1)\}$;
- At $(1,1)$, $f(k,t) \to 0^+$, causing $\frac{1}{\sqrt{f(k,t)}} \to +\infty$, and thus $\frac{\partial L_{TN}(k,t)}{\partial t} \to -\infty$.

Hence, $\frac{\partial L_{TN}(k,t)}{\partial t}$ has a *single isolated infinite discontinuity* at $(k,t) = (1,1)$, and is strictly negative everywhere else in the domain:

$$\frac{\partial L_{TN}(k,t)}{\partial t} < 0, \quad \forall (k,t) \in [0,+\infty) \times [-1,1] \setminus \{(1,1)\}. \tag{A-3}$$

This implies that, for any fixed $k > 0$ with $k \neq 1$, the function $L_{TN}(k,t)$ is strictly decreasing in $t$ over $[-1,1)$. Although the derivative is undefined at $(1,1)$, we verify that $L_{TN}(k,t)$ itself remains continuous at this point (by direct substitution into the original loss function). Therefore, the minimum value of $L_{TN}(k,t)$ with respect to $t$ occurs at the right endpoint $t = 1$, for all $k > 0$.

For all $k > 0$, the function $L_{TN}(k,t)$ attains its global minimum over $t \in [-1,1]$ at $t = 1$, despite the singularity in the derivative at $(k,t) = (1,1)$.

The discontinuity in the derivative does not affect the existence or location of the minimum because the function $L_{TN}(k,t)$ is continuous on the closed domain $[0,+\infty) \times [-1,1]$. The monotonicity holds almost everywhere, and the endpoint $t = 1$ remains the unique minimizer by continuity and boundary analysis.

This result justifies our choice of $t = 1$ as the optimal setting in the training objective, ensuring stability and convergence properties in the optimization landscape.

## B    DIFFERENCES BETWEEN REPRODUCED RESULTS OF VLM2VEC-V2 FOR FULL TASKS AND THE ORIGINAL PAPER

We first acknowledge the open-source release of VLM2Vec-V2. In the original VLM2Vec-V2 paper, the authors report an average performance of 65.4 on the Hit@1 metric for the VisDoc task across the full-task training set. However, in our work, we reproduce the results using the same hardware configuration as the original paper and strictly follow its open-source code with default hyperparameters. Our results are only 61.1[6], significantly lower than the original reported value, multiple issues in the official code repository report similar reproduction failures[7][8]. Currently, we and our peers preliminarily speculate that this inconsistency may stem from version differences in the DATASETS package. There may be implicit behavioral changes in data loading, sampling order, or preprocessing workflows across different versions of the datasets package[9], which could affect the stability of model training and evaluation. However, as of the submission of this paper, the official repository for VLM2Vec-V2 still does not explicitly specify the exact versions of its dependencies. To ensure fairness and comparability in experimental evaluation, we still adopt the currently reproducible baseline result as the comparison baseline. Our method achieves superior performance under identical training and evaluation environments, leading to a reasonable inference: compared to the current implementation of VLM2Vec-V2, our approach inherently demonstrates greater effectiveness and robustness. We report the results of the original VLM2Vec-V2 paper and our reproduction in Table B-1 and compare them with our proposed TNCME. We commit to retraining and reevaluating our method in an environment that can reproduce the original results, and to reporting the results in our open-source repository if this reproducibility issue is resolved in the future. We once again sincerely thank the VLM2Vec-V2 team for their valuable contributions to the open-source community.

## C    TRAINING AND EVALUATION DATASETS

VLM2Vec-V2 constructs a training-evaluation framework for multimodal retrieval tasks called MMEB-V2. The training dataset comprises three categories: image-text, VisDoc-text, and video-text retrieval data. The following table details the sub-datasets and quantities within each dataset category. The MMEB-V2 benchmark test set comprises 81 sub-tasks across three major retrieval task categories, organized under nine meta-tasks. These cover the three primary modalities: images, videos, and visual documents. Completely independent of the training set, this test set measures the model's generalization capabilities. Tables C-1 and C-2 detail the sources and quantities for each task. To uniformly evaluate model performance across multimodal retrieval tasks, we use the average performance across each modality's test set as the evaluation metric.

## D    EVALUATION METRICS

In this section, we summarize the evaluation metrics employed in the task in Table D-1.

---

[6]The benchmark MMEB-V2, introduced by VLM2Vec-V2, actually encompasses 27 VisDoc tasks. However, VLM2Vec-V2 evaluated only 24 of them, excluding three multilingual tasks. In our evaluation, we have tested all 27 VisDoc tasks.

[7]https://github.com/TIGER-AI-Lab/VLM2Vec/issues/130

[8]https://github.com/TIGER-AI-Lab/VLM2Vec/issues/149

[9]To avoid breaking double-blind protocols, this is the outcome of our discussions conducted through alternative communication channels rather than via issues in the VLM2Vec-V2 official repository.

Table B-1: We report the original results of VLM2Vec-V2 in this table, including several baselines. We also report our reproduced results for VLM2Vec-V2 under identical conditions, along with the results for TNCME. The evaluation metrics are consistent with the VLM2Vec-V2.

| Model | Image (Hit@1) 36 Avg. | Video (Hit@1) 18 Avg. | VisDoc (NDCG@5) 24 Avg. | All | Train Sets |
|---|---|---|---|---|---|
| ColPali-v1.3 | 34.9 | 28.2 | 71.0 | 44.5 | |
| GME-2B | 51.9 | 33.9 | 72.7 | 54.1 | |
| GME-7B | 56.0 | **38.6** | **75.2** | **57.9** | Not Reported in VLM2Vec-V2 |
| LamRA-Qwen2-7B | 54.1 | 35.2 | 23.9 | 40.4 | |
| LamRA-Qwen2.5-7B | 52.4 | 33.7 | 50.2 | 47.4 | |
| VLM2Vec-Qwen2VL-2B | 59.7 | 29.0 | 41.6 | 47.0 | |
| VLM2Vec-Qwen2VL-7B | **65.5** | 34.0 | 46.4 | 52.4 | |
| VLM2Vec-V2-2B Reported 5k steps | **64.9** | **34.9** | **65.4** | **58.1** | All Sets |
| VLM2Vec-V2-2B Reproduced 5k steps | 64.4 | 33.4 | 61.1 | 56.2 | |
| VLM2Vec-V2-2B Reproduced 2k steps | 62.4 | 35.0 | 57.9 | 54.7 | All Sets |
| TNCME-2B 2k steps | 62.8 | 35.2 | 58.1 | 55.0 | |
| VLM2Vec-V2-2B Reproduced 5k steps | 63.7 | 30.0 | 29.9 | 45.5 | Image-Text Only |
| TNCME-2B 2k steps | 64.9 | 31.4 | 31.2 | 46.8 | |
| VLM2Vec-V2-2B Reproduced 5k steps | 64.5 | 32.9 | 56.8 | 54.8 | Image-Text & VisDoc-Text |
| TNCME-2B 2k steps | **65.3** | 33.0 | 57.9 | 55.6 | |

Table C-1: Sub-datasets for the Image-Text retrieval tasks and VisDoc-Text retrieval tasks in MMEB-V2.

| Task Type | Task Name | Reference |
|---|---|---|
| Visual Question Answering | OK-VQA | (Marino et al., 2019) |
| | A-OKVQA | (Schwenk et al., 2022) |
| | DocVQA | (Mathew et al., 2021) |
| | InfoVQA | (Mathew et al., 2022) |
| | ChartQA | (Masry et al., 2022) |
| | Visual7W | (Zhu et al., 2016) |
| | ScienceQA | (Lu et al., 2022) |
| | GQA | (Hudson & Manning, 2019) |
| | TextVQA | (Singh et al., 2019) |
| | VizWiz | (Gurari et al., 2018) |
| Image Classification | Voc2007 | (Everingham et al., 2010) |
| | N24News | (Wang et al., 2022) |
| | SUN397 | (Xiao et al., 2010) |
| | ObjectNet | (Barbu et al., 2019) |
| | Country211 | (Radford et al., 2021b) |
| | Place365 | (Zhou et al., 2018a) |
| | ImageNet-1K | (Russakovsky et al., 2015) |
| | ImageNet-A | (Hendrycks et al., 2019) |
| | ImageNet-R | (Hendrycks et al., 2021) |
| | HatefulMemes | (Kiela et al., 2020) |
| Image-level Retrieval | MSCOCO_I2T | (Lin et al., 2014) |
| | MSCOCO_T2I | (Lin et al., 2014) |
| | VisDial | (Das et al., 2017) |
| | CIRR | (Liu et al., 2021b) |
| | VisualNews I2T | (Liu et al., 2021a) |
| | VisualNews T2I | (Liu et al., 2021a) |
| | NIGHTS | (Diament et al., 2023) |
| | WebQA | (Chang et al., 2022) |
| | EDIS | (Liu et al., 2023b) |
| | OVEN | (Hu et al., 2023) |
| | WIKI-SS-NQ | (Ma et al., 2024a) |
| | FashionIQ | (Wu et al., 2021) |
| Visual Document Retrieval | ViDoRe | (Faysse et al., 2025b) |
| | ViDoRe-v2 | (Macé et al., 2025) |
| | VisRAG | (Yu et al., 2025) |
| | ViDoSeek | (Wang et al., 2025a) |
| | MMLongBench-Doc | (Ma et al., 2024b) |
| Visual Grounding | MSCOCO | (Lin et al., 2014) |
| | RefCOCO | (Kazemzadeh et al., 2014) |
| | RefCOCO-Matching | - |
| | Visual7W-Pointing | (Zhu et al., 2016) |

Table C-2: Sub-datasets for the VisDoc-Text retrieval tasks in MMEB-V2.

| Task Type | Task Name | Reference |
|---|---|---|
| Video Question Answering | Video-MME | (Fu et al., 2025) |
| | MVBench | (Li et al., 2024) |
| | NExT-QA | (Xiao et al., 2021) |
| | EgoSchema | (Mangalam et al., 2023) |
| | ActivityNetQA | (Yu et al., 2019) |
| Video Classification | UCF101 | (Soomro et al., 2012) |
| | HMDB51 | (Kuehne et al., 2011) |
| | Kinetics-700 | (Carreira et al., 2019) |
| | Breakfast | (Kuehne et al., 2014) |
| | Something-Something V2 | (Goyal et al., 2017) |
| Video-level Retrieval | MSR-VTT | (Xu et al., 2016) |
| | MSVD | (Chen & Dolan, 2011) |
| | DiDeMo | (Hendricks et al., 2017) |
| | VATEX | (Wang et al., 2019) |
| | YouCook2 | (Zhou et al., 2018b) |
| Moment Retrieval | QVHighlights | (Lei et al., 2021) |
| | Charades-STA | (Gao et al., 2017) |
| | MomentSeeker | (Yuan et al., 2025) |

Table D-1: Evaluation Metrics and Their Meanings

| Metric | Description |
|---|---|
| Hit@K | Proportion of queries where the correct item is ranked within top-K. |
| NDCG@K | Normalized ranking quality; supports linear (0/1) or exponential ($2^{rel} - 1$) relevance gain. |
| Precision@K | Fraction of retrieved top-K items that are relevant. |
| Recall@K | Fraction of all relevant items retrieved in top-K. |
| F1@K | Harmonic mean of Precision@K and Recall@K. |
| MAP@K | Mean of Average Precision across queries, truncated at rank K. |
| MRR@K | Mean reciprocal rank of the first relevant item (capped at K). |

