# OpenReview forum: "TNCME: Tensor's Norm Constraints for Unsupervised Contrastive Learning of Multimodal Embeddings"
_ICLR.cc/2026/Conference — ICLR 2026 Conference Withdrawn Submission_

### Official Review · Reviewer_Tr6q · 2025-10-15

**Soundness:** 3
**Presentation:** 2
**Contribution:** 3
**Rating:** 4
**Confidence:** 5

**Summary:**

This paper focuses on two-modal learning and proposes a reframe of the classical paradigm of contrastive learning in which embeddings are normalized to unitary norm and then put closer or apart according to the cosine similarity. The authors propose to also leverage the magnitude of embeddings to build more comprehensive representations. The idea sounds and it is interesting to propose something that goes beyond standard contrastive learning approaches.

**Strengths:**

The idea of considering the embedding magnitude too sounds and represents an advancement wrt conventional contrastive learning approaches.

The analysis of losses global minimum is thorough and it well supports the theoretical definitions.

Results, although very confusing, are convincing.

**Weaknesses:**

W1) The whole analysis is based on the assumption that we can set ||h_tgt|| = k \cdot ||h_qry||, but no theoretical or empirical justification is provided for this assumption.

W2) The inequality 4 is not explained but it is crucial for the rest of the method.

W3) The paper is an extension of TNCSE, which decreases a little bit the novelty. Also, this paper follows the same exact structure of TNCSE. Nevertheless, I still think that the paper is relevant for the multimodal learning community.

W4) (Less important wrt to previous ones, but still a wekness) The authors should better explain the rationale behind the temperature choice. Why tau_TN is set to an extremely low value 10^-4 for full training set and to higher values 0.05 in subset-based? It is well-known that the temperature parameter has a strong impact on performance and especially on the contributions of hard negatives in contrastive Learning.

**Questions:**

Q1) why do we set ||h_tgt|| = k \cdot ||h_qry||? Can the authors provide justification for this assumption?
Q2) Can the authors better explain how inequality 4 is constructed? which are the Mathematical passages that prove it?
Q3) I would be curious to see the plots for the conventional cosine-only loss minimum.
Q4) It would be interesting to plot also the standard InfoNCE loss in the plot in fig5b to verify if the smoother behavior of the blue loss is only a contribution of the infoTN or of the infoNCE loss structure.
Q5) In the trend chart of fig.2 the behavior of L_TN seems smooth. So, why does it produce such a bad loss landscape in fig5?
Q6) The average Euclidean distance in fig.4 is computed on the reduced-space tsne space or on the original multimodal space? Also, can the authors provide both the euclidean and cosine distance in the original space?

---

### Official Review · Reviewer_irPe · 2025-10-31

**Soundness:** 2
**Presentation:** 1
**Contribution:** 1
**Rating:** 2
**Confidence:** 3

**Summary:**

The paper proposes TNCME, a framework that jointly optimizes the angle and magnitude of multimodal embeddings. To demonstrate the effectiveness of TNCME, the paper reproduce VLM2Vec with both TNCME and (VLM2Vec's) InfoCE loss. TNCME is validated against the baseline on Image-to-text, visual document and video-text retrieval.

**Strengths:**

- To the reviewer’s knowledge, the work is the first to investigate using unnormalized multimodal embeddings.

**Weaknesses:**

- The main contribution of the work seems to be a relatively straightforward extension of [1] into the multimodal setting, by applying softmax to $L_{TN}$. As is, the reviewer has concerns about the novelty of the work.

- Evaluation of the loss function is limited. In particular, evaluation of dual encoder architectures would be interesting.

- Especially in light of the reproduction issues, evaluating on top of multiple baselines (such as [2]) would be beneficial. Especially as GME seems to outperform the reproduction of VLM2VEC.

- The motivation behind the analysis in section 3.1/3.2 is unclear (contribution 2). From equation 1, it is obvious that $L_{TN}$ is minimized when $h_{qry} = h_{tgt}$. It is unclear why the analysis in the rest of the section is required.

- The authors conclude that the instability of using $L_{TN}$ directly is “due to the batch size.” However, this is not justified empirically or theoretically.

- The motivation for figure 4 is unclear. As VLM2Vec does not optimize euclidean distance it is unsurprising that it has larger euclidean distance between pairs than TNCME. It is also unclear how this relates to “cross-modal retrieval alignment capabilities”

- The manuscript is confusing to read, and introduces some unnecessary jargon.

minor:

- Inconsistent spacing between text and citations.

**Questions:**

Some questions can be found in the above weaknesses section, in addition:

- How sensitive is the proposed loss function to the setting of lambda?

---



[1] T. Zong, B. Shi, H. Yi, and J. Xu, “TNCSE: Tensor’s Norm Constraints for Unsupervised Contrastive Learning of Sentence Embeddings,” in Proceedings of the AAAI Conference on Artificial Intelligence, 2025, pp. 26192–26201. doi: 10.1609/aaai.v39i24.34816.


[2] X. Zhang et al., “GME: Improving Universal Multimodal Retrieval by Multimodal LLMs,” 2024, doi: 10.48550/arxiv.2412.16855.

---

### Official Review · Reviewer_BLr7 · 2025-11-01

**Soundness:** 1
**Presentation:** 3
**Contribution:** 2
**Rating:** 2
**Confidence:** 4

**Summary:**

This paper discusses cross-modal retrieval using embedded representations of content in different modalities.  The embeddings used in such tasks are typically normalized to be of unit norm, thereby their inner product represents the directional alignment (cosine of angle) between the two vectors.  This paper argues for comparing the norm of the embedding vectors as well, as was done in an earlier paper for text only embeddings (a method termed TNCSE).  This paper applies TNCSE to cross-modal retrieval, developing a method termed TNCME.  Paper also shows that the proposed tensor norm based objective is monotonic with a unique, global optima with respect to two parameters: the ratio of norms of embeddings being compared, and cosine of angle between them. Empirical results are presented on image-text, visdoc-text, and video-text retrieval tasks.

**Strengths:**

* A novel model and training objective that aims to include norm of embedding vectors in addition to their directional alignment for multimodal retrieval tasks.
* Proposed approach shows a small but consistent gain on image-text, visdoc-text, and video-text retrieval tasks.

**Weaknesses:**

* The proposed objective of TN or InfoTN is inclusive of norm and direction between embeddings, then why is it still mixed with InfoNCE which is based on direction only and should be subsumed by InfoTN?
* Furthermore, once the model is trained, why is the output of NAP layer not the only output used for retrieval?  These embeddings have been trained for both norm and directional alignment and would be expected to be better?
* NAP module adds extra parameters to the model, making it unclear if the gains truly come from the proposed norm-alignment objective or simply from extra parameters.  Is it possible to do a fairer comparison?  For instance, how would InfoNCE on normalized output of NAP perform?
* Since the main argument of the paper is use of norm in addition to angle between embedding vectors, it is surprising to see that the using norm with original embedding (that is using embedding before the it undergoes RMSNorm step), as discussed in Section 5.3, does not result in a good model despite training.
* Theoretical analysis demonstrates that the tensor norm objective is monotonic with unique global minima.  However this is only as a function of two parameters — the ratio of norms of the two vectors being compared and cosine of angle between them.  How this function behaves as a function of model parameters being optimized is not discussed.

**Questions:**

n/a

---

### Official Review · Reviewer_whas · 2025-11-02

**Soundness:** 3
**Presentation:** 3
**Contribution:** 2
**Rating:** 4
**Confidence:** 3

**Summary:**

The paper targets norm imbalance in multimodal embeddings and proposes a training-time norm alignment approach that couples a Tensor Norm constraint with InfoNCE (InfoTN) plus a lightweight Norm Alignment Projector used only during training. This recipe aims to be plug-in for existing backbones, reports consistent top-1 retrieval gains across many benchmarks with no added inference cost, and includes analyses on stability and hyperparameters.

**Strengths:**

The paper is original in jointly aligning direction and magnitude by treating the L2 norm as semantic signal, introducing InfoTN for cross-modal norm mismatch, and adding a Norm Alignment Projector. Quality is strong with a clear optimality analysis of the LTN objective and extensive experiments across 81 retrieval tasks with thorough ablations and ranking metrics. Clarity is high thanks to intuitive figures, clean derivations, and reproducible training details. Significance is evident in consistent Top-1 gains over strong baselines, robustness across modalities and budgets, and a simple plug-and-play design that can impact future multimodal retrieval.

**Weaknesses:**

The novelty is limited compared with existing work. Evaluation is narrowly tuned to Top-1; add significance tests, broader metrics (recall at k, calibration), and per-task deltas with confidence intervals. Training appears hyperparameter sensitive. Clarify the train–test mismatch from dropping the projector at inference and test more backbones and sizes to demonstrate robustness.

**Questions:**

Could you report significance tests and a broader metric suite beyond Top-1, including recall at k and calibration, with per-task deltas and confidence intervals; clarify why your replicated baseline underperforms and, if possible, release verified checkpoints or reproduce with third-party weights; provide sensitivity analyses over temperatures, loss weights, batch size, and modality balance, along with documented failure modes; run an ablation that keeps a lightweight alignment at inference to address the potential train-test mismatch from dropping the projector; evaluate on multiple backbones and sizes to show transferability; and share training cost and stability characteristics so we can judge practicality at scale?

---

### Note · Authors · 2025-11-12

I have read and agree with the venue's withdrawal policy on behalf of myself and my co-authors.